# The Microbiome as a Therapy in Pouchitis and Ulcerative Colitis

**DOI:** 10.3390/nu13061780

**Published:** 2021-05-23

**Authors:** Jean-Frédéric LeBlanc, Jonathan P. Segal, Lucia Maria de Campos Braz, Ailsa L. Hart

**Affiliations:** 1Inflammatory Bowel Disease Unit, St. Mark’s Hospital, Harrow HA1 3UJ, UK; lucia.braz@nhs.net (L.M.d.C.B.); ailsa.hart@nhs.net (A.L.H.); 2Department of Gastroenterology, The Hillingdon Hospital, Uxbridge UB8 3NN, UK; jonathansegal1@nhs.net; 3Faculty of Medicine, Department of Metabolism, Digestion and Reproduction, Imperial College, London SW7 2AZ, UK

**Keywords:** ulcerative colitis, ileoanal pouch inflammatory bowel disease, microbiota

## Abstract

The gut microbiome has been implicated in a range of diseases and there is a rapidly growing understanding of this ecosystem’s importance in inflammatory bowel disease. We are yet to identify a single microbe that causes either ulcerative colitis (UC) or pouchitis, however, reduced microbiome diversity is increasingly recognised in active UC. Manipulating the gut microbiome through dietary interventions, prebiotic and probiotic compounds and faecal microbiota transplantation may expand the therapeutic landscape in UC. Specific diets, such as the Mediterranean diet or diet rich in omega-3 fatty acids, may reduce intestinal inflammation or potentially reduce the risk of incident UC. This review summarises our knowledge of gut microbiome therapies in UC and pouchitis.

## 1. Introduction

The human gut microbiome represents the collective genomes of a vast range of microorganisms, also referred to as the microbiota, which include bacteria, virus, archaea and protozoa, that together form an extremely complex ecosystem capable of communicating with the immune system and determining an individual’s predisposition to develop disease states. Gut microbiota are responsible for a range of metabolic processes for the host including digesting food specifically through the breakdown of complex carbohydrates and proteins, regulation of the immune function, protecting against pathogens and production of micronutrients [1].

The gut microbiome composition is determined by a number of host and environmental factors including mode of delivery, type of feeding at birth, use of antibiotics, pathogen exposure, diet, smoking, pollutants as well as many unknown factors [2]. Specific bacterial phyla, such as Firmicutes and Bacteroides, represent approximately 90% of the human gut microbiota, alongside smaller proportions of Proteobacteria, Actinobacteria and Verrucomicrobia [3,4]. Dysbiosis is most commonly defined as perturbations in the composition of commensal communities relatively to those found in healthy individuals. For example, an increased Firmicutes/Bacteroidetes ratio may be associated with disease states, such as obesity, type 2 diabetes and metabolic syndrome [5]. Reduced Firmicutes diversity, especially reduced levels of *F. prausnitzii*, was observed in the gut microbiota of patients with inflammatory bowel disease (IBD) [6]. 

Inflammatory bowel disease represents a group of chronic intestinal disorders, subdivided into Crohn’s disease (CD) and ulcerative colitis (UC). Bowel inflammation, responsible for symptoms of abdominal pain and bloody diarrhoea in UC, may be partly mediated by changes in the composition of the gut microbiota. UC currently affects 10 per 100,000 people annually, with a prevalence of 243 per 100,000, thus amounting to approximately 146,000 patients diagnosed with UC in the UK [7]. Similarly to other autoimmune diseases, the precise aetiology of UC has not been determined and is likely multifactorial, involving complex interactions between the genome and epigenome, the environment and the microbiome [2]. Patients with moderate-to-severe UC can be treated with biological agents, whose immunosuppressive effects strive to achieve the endpoint of mucosal healing of the bowel. However, pooled rates of mucosal healing in UC (33–45%) were relatively modest, combined with non-negligible rates of adverse events, as observed in a network meta-analysis of clinical trials [8]. Therapies involving manipulation of the microbiota may thus expand the therapeutic arsenal of UC. 

When trying to understand the role of the gut microbiome on the aetiology of UC, longitudinal data are lacking. A particular challenge remains that we still are unable to predict those that will develop UC and therefore we lack the ability to map the microbiome prior to the disease onset. In an attempt to circumnavigate this issue, a model for UC may provide us unique insights into the role of the gut microbiome in IBD. In such a model, the key will be to explore microbiome changes from a period of health to one of disease state, mapping the microbiome changes. Ideally, these changes need to occur over a period where longitudinal data is possible, minimising loss to follow-up.

A potential model for understanding dysbiosis within different UC phenotypes is the ileal pouch-anal anastomosis (IPAA), which is a surgically constructed intestinal reservoir connecting the most distal part of the small bowel to the anal canal. Total proctocolectomy may be indicated in patients who remain refractory to medical therapy or those at high risk of colorectal malignancy; in order to restore intestinal continuity and avoid a permanent ileostomy, an IPAA can be formed. Inflammation of the IPAA, also called pouchitis, is a relatively frequent complication, as shown by a cumulative risk of one or more episodes ranging from 15–53% [9]. Incidence of pouchitis seems highest within the first year of pouch formation, allowing longitudinal data in a short timeframe. Observed predictors of pouchitis include prior smoker status, associated extra-intestinal manifestations, primary sclerosing cholangitis and a genetic marker called interleukin-1 receptor antagonist gene allele 2 [10]. The mainstay of treatment for pouchitis is antibiotics and many studies have implicated the importance of the microbiome in both aetiology and treatment.

This review aims to summarise the potential role of the microbiome as a therapeutic target in both UC and pouchitis.

### 1.1. Interactions between the Gut Microbiome and the Intestinal Lining in the Context of Intestinal Inflammation

The gut microbiome seems an integral part in maintaining the tight junction integrity [11]. Evidence has suggested that perturbations in the gut microbiome can lead to an increase in gut permeability, a decrease in the thickness of the protective mucus layer which culminates in pathogen invasion [12]. The goblet cells, found in the intestinal lining, produce a thick mucus layer which acts as a mechanical barrier, as well as chemical, in that it supports the presence of antibacterial proteins, such as secretory IgA and lactoferrin [13,14]. Therefore, a loss of goblet cells, as found in chronic bowel inflammation, leads to a loss in mucosal integrity and impaired barrier function. This culminates to an increase in bacterial translocation altering the T-cell profile and pro-inflammatory cytokines leading to tissue damage, a destabilised microbiome and further tissue damage. Short-chain fatty acids (SCFA), mainly butyrate, are produced via selective bacterial fermentation of resistant carbohydrates. SCFA play an important role in the maintenance of the epithelial barrier and regulation of the immune system, through increased intestinal IgA production, induction of tolerogenic dendritic cells and higher regulatory T-cell percentages [15]. Indeed, bowel inflammation has been associated with increased rates of intestinal regulatory T lymphocytes, however it remains unclear which subsets of regulatory T cell exert proinflammatory effects in UC [16].

From an immune perspective, the gut microbiome is in constant communication with the immune system to help with immune tolerance and disease pathogenesis [17]. Breakdown of the epithelial innate immune function may also lead to the invasion of the intestinal lining by pathogenic bacteria. Indeed, defective autophagy impairs lysosomal digestion and clearance of pathogenic bacterial strains. Activation of different signalling cascades, mediated by the Toll-like receptors (TLR) located on the surface of intestinal epithelial cells and the downstream nuclear factor-κB (NF-κB) pathway, leads to the production of cytokines and recruitment of the adaptive immune system, thus enhancing bacterial clearance. Thus, genetic polymorphisms of TLR, inhibition of the NF-kB pathway and decreased release α-defensin seen in variants of the caspase recruitment domain/nucleotide-binding oligomerisation domain (CARD/NOD family) all contribute to increased intestinal permeability and subsequent bowel inflammation, as seen in UC [18]. 

Through advancing next generation sequencing technologies, we have been able to understand both the composition of the gut microbiota and also its functionality [19]. Figure 1 summarises the critical roles of the microbiome in drug metabolism, absorptive and secretory capacities of the intestinal lining. Although data remain heterogenous, consistent findings of perturbations in the gut microbiota in patients with UC compared with healthy controls persist. In an inception cohort, Schirmer et al. evaluated the microbial taxonomic changes of 405 paediatric new-onset, treatment-naïve UC patients, who received either corticosteroids or 5-aminosalicylic acid drugs. Interestingly, after 52 weeks of follow-up, poor response to therapy correlated with an increased resemblance to the bacterial taxa of the oral cavity, such as higher levels of *H. parainfluenzae*, likely explained by intestinal inflammation and strain-specific adaptation [20]. However, it remains unclear whether such microbial changes may be the cause or the consequence of active bowel inflammation.

Interestingly, microbial signatures may differ between UC and CD despite their similar clinical and epidemiologic profiles. In a Spanish cohort of 34 patients with CD and 33 with UC, those afflicted with CD showed, in faecal samples, higher rates of unstable microbial communities compared to patients with UC, as well as increasingly altered microbiota composition, such as a lower relative abundance of *Faecalibacterium* and a higher relative abundance of *Fusobacterium* and *Escherichia*. The composite microbial biomarkers were validated as a diagnostic tool in Spanish and Belgian cohorts, demonstrating a sensitivity of 80% and a specificity of 90.9% in distinguishing CD from UC [21].

### 1.2. Influences of the Gut Microbiota on the Development of Pouchitis

The microbiota of a patient with pouchitis differs from a patient without pouchitis. Specific patterns have found persistence of Fusobacter and Enterobacters associated with the disease state and the absence of specific bacteria such as *Streptococcus* species in the inflamed pouch [22]. Clinically, pouchitis can be treated with antibiotics and hence provides plausibility that the microbiome may influence the course of pouchitis. Of interest, mucosal inflammation in the pouch is concentrated in areas where bacterial concentration is highest [23]. From an immune perspective, it has been noted that pouchitis-derived bacterial sonicates from metronidazole-sensitive bacterial species stimulate healthy patients’ mononuclear cells significantly more than corresponding sonicates from non-pouchitis patients [24].

Potentially, the closest direct link to the microbiome having an influence on pouchitis came from a study which highlighted that baseline microbiome prior to colectomy could predict those that developed pouchitis and those that did not and hence provides suggestions that altering the gut microbiota may influence the pouch functionality [25].

## 2. Selective Therapies Aiming at Microbiota Manipulation in Ulcerative Colitis

The gut microbiota can be manipulated through the use of prebiotics, probiotics, antibiotics and faecal microbiota transplantation and diet (Figure 2).

### 2.1. Prebiotic Studies in Ulcerative Colitis

Dietary prebiotics are defined as ‘a substrate that is selectively utilised by host micro-organisms conferring a health benefit’ [26]. It is hypothesised that prebiotics may improve gut inflammation by selective stimulation of protective members of gut microbiota, improvement of the intestinal permeability and increased production of SCFA [27].

Hafer et al. studied the effect of lactulose at a daily dose of 10 g added to standard therapy in 7 patients with active UC, compared to 7 UC patients receiving standard therapy without the use of a placebo. They noted that the Inflammatory Bowel Disease Questionnaire (IBDQ) score improved from 123 ± 20 to 171 ± 18 (*p* = 0.026) in the lactulose group [28].

Two clinical trials evaluated the effects of an oligofructose-enriched inulin compound (Beneo™ Synergy 1) in UC patients with mild to moderate disease, receiving concomitant mesalazine therapy. Casellas et al. noted that, at day 14, levels of faecal calprotectin improved in five of seven patients (70%) who received the prebiotic, compared to two of eight patients (25%) in the placebo arm [29].

Valcheva et al. assessed the alterations of the gut microbiota composition and activity in 25 patients with mild to moderate UC treated with different doses of oligofructose-enriched inulin over a nine-week period. The primary outcome was clinical response and/or remission. The primary outcome was achieved in 77% of patients receiving the high-dose prebiotic product (15 g per day) compared to 33% in the low-dose group (7.5 g per day). High-fructan dose was associated with *Bifidobacteriaceae* and *Lachnospiraceae* abundance, however such microbiota modifications did not correlate with improved disease scores. Interestingly, the trial showed that a prebiotic course resulted in higher butyrate levels, with strong negative correlations between butyrate levels and clinical symptoms [30]. Existing trials assessing the use of prebiotics in the treatment of UC lack sufficient power to change clinical practice, however data regarding its potential efficacy and safety profile are encouraging.

### 2.2. Probiotic Studies in Ulcerative Colitis

Probiotics are ‘live microorganisms which confer a health benefit on the host when administered in adequate amounts’ [26]. Probiotics are traditionally composed of one or more bacterial strains.

Derwa et al. performed a meta-analysis of eight trials targeting induction of remission in active UC as a primary outcome (*n* = 651), as well as six trials assessing prevention of relapse in quiescent UC (*n* = 677) [31]. Types of probiotics varied between *E. coli* Nissle 1917 (5 studies), *Bifidobacterium* longum 356 (1 study), Lactobacillus rhamnosus GG (1 study), a multistrain probiotic containing a combination of lactic acid bacteria, streptococci and bifidobacterial (3 studies) and other combined formulations (4 studies). In the single trial comparing probiotics with 5-ASAs for induction of remission in 116 patients, no difference was seen in the primary outcome of failure to achieve remission (RR = 1.24; 95% CI = 0.70–2.22), similar to the pooled 7 randomised placebo-controlled trials (RR = 0.86; 95% CI = 0.68–1.08). Rates of adverse events were comparable in both analyses. Interestingly, in a subgroup analysis of the multistrain probiotic containing a combination of lactic acid bacteria, streptococci and bifidobacteria studies, 56.2% of 162 patients randomised to the probiotic failed to achieve remission, compared with 75.2% of 157 patients who received placebo (RR = 0.74; 95% CI = 0.63–0.87). The authors measured a number needed to treat of 5 to prevent one patient with active UC failing to achieve remission, without significant heterogeneity between studies (I^2^ = 0%, *p* = 0.52). In a pooled group of 140 patients, the *E. coli* Nissle 1917 compound did not demonstrate a statistically different benefit compared to placebo (RR = 1.56; 95% CI = 0.44–5.53). In a randomised, double-blind trial, seven weeks of E. coli Nissle 1917 or placebo was added to conventional therapy in patients with active UC and initially treated with one week of ciprofloxacin or placebo (25 patients in each of the four groups). Surprisingly, rates of clinical remission and treatment persistence with the study drug were significantly lower in patients treated with E. coli Nissle 1917 and without a previous antibiotic cure, despite similar adverse events in all groups [32]. In regard to maintenance of remission, use of probiotics in 342 patients was not shown to decrease rates of UC relapse compared with the use of 5-ASAs (RR = 1.02; 95% CI = 0.85–1.23) in 278 patients and placebo (RR = 0.62; 95% CI = 0.33–1.16) in 57 patients.

Astó et al. conducted a meta-analysis of randomised controlled trials (RCTs) examining the effects of probiotics, prebiotics and synbiotics on human UC [33]. Rates of remission in 602 patients with active UC were unchanged between the probiotics and placebo groups. In trials defining UC remission, the beneficial effects of probiotics were estimated to be statistically significant compared to placebo (RR = 1.55, 95% CI = 1.13–2.15) with decreased heterogeneity between trials (I^2^ = 29%). On further subgroup analysis (n = 424), patients with active UC who received *Bifidobacterium*-containing probiotics were more likely to be in remission compared to those on placebo (RR = 1.73; 95% CI = 1.23–2.43, *p* = 0.002). In comparison, no difference in UC remission was seen between probiotics without *Bifidobacterium* strains and control groups (*n* = 168). In trials assessing the multistrain probiotic containing a combination of lactic acid bacteria, streptococci and bifidobacteria in combination with standard therapy (*n* = 348), significantly higher rates of UC remission were seen in the probiotic group compared to the control group (RR = 1.99; 95% CI = 1.25–3.15, *p* = 0.003).

The faecal concentrations of SCFA were measured in two trials and were significantly increased in one pilot study with active UC patients after supplementation of *Bifidobacterium*-fermented milk (*n* = 20) [34]. In a trial assessing 46 inactive UC patients, SCFA concentrations did not differ significantly between probiotic (*Streptococcus faecalis* T-110, *Clostridium butyricum* TO-A and *Bacillus mesentericus* TO-A) and placebo groups at any time over the six months, however a higher butyrate/acetate ratio was observed throughout the follow-up period in patients who relapsed compared to those who remained in remission [35]. Decreased *Bifidobacterium* species was observed in 195 inactive UC patients prior to relapse in one study assessing the effects of *Bifidobacterium breve* fermented milk [36]. Furrie et al. explored the effects of a synbiotic formulation containing *B. longum* and oligofructose-enriched inulin; improved endoscopic scores and significantly higher levels of bifidobacterial rRNA on mucosal biopsies were observed in 9 patients receiving this synbiotic compared to 9 patients on placebo [37].

### 2.3. Antibiotics in Ulcerative Colitis

Antibiotics are seldom used in clinical practice for the management of UC. A few initial studies suggested a potential clinical benefit of adding antibiotics, such as ciprofloxacin or tobramycin, to conventional therapy in patients with active UC of all severities [38,39]. Khan et al. performed a meta-analysis of nine RCTs assessing the efficacy of antibiotics in adult patients with active UC [40]. Efficacy outcomes were mostly clinical with limited reporting of biochemical and endoscopic outcomes. Overall, the authors noted a statistically significant benefit favouring antibiotics over placebo (RR 0.64; 95% CI = 0.43–0.96, *p* = 0.03), however antibiotic regimens were significantly heterogeneous between trials. Two RCTs assessed the use of antibiotics in acute severe UC, a condition perceived at high risk of bacterial translocation, and showed no short-term clinical benefit of adding metronidazole (trial of 39 patients) or metronidazole and tobramycin (trial of 39 patients) compared to placebo [41,42]. Internationally recognised guidelines either do not recommend their use or do not mention them as a potential therapeutic option in the management of adult patients with UC [43,44].

### 2.4. Faecal Microbiota Transplantation in Ulcerative Colitis

Since 2015, four placebo-controlled RCTs [45,46,47,48] and multiple cohort studies have been published [49,50,51], with meta-analyses suggesting a positive impact of faecal microbiota transplantation (FMT) in the induction of remission in UC patients with mild-moderate disease [52].

Costello et al. conducted a meta-analysis of the four placebo-controlled RCTs, totalling 277 patients. Clinical remission was achieved in 28% of pooled donor FMT groups compared with 9% of patients in placebo groups (OR = 3.67; 95% CI = 1.82–7.39) [53]. A Danish open-label pilot study has examined the efficacy of oral FMT capsules in patients with active UC; over a 50-day course of oral FMT capsules, all of the seven patients achieved clinical response at weeks 4 and 8, as well as significant improvements in quality of life and faecal calprotectin levels [49].

Paramsothy et al. performed gastrointestinal microbial community profiling in 81 UC patients treated with colonoscopy delivered FMT versus placebo. Bacterial diversity in samples before and after FMT administration was higher in recipients who achieved remission compared with those who did not. Remission after FMT was associated with a relative microbial abundance of *Eubacterium hallii* and *Roseburia inulivorans*, compared with higher levels of *Fusobacterium gonidiaformans*, *Sutterella wadsworthensis* and *Escherichia* species in patients not determined to be in remission post-FMT. The former patient group exhibited increased production of SCFA and secondary bile acids, while the latter group showed higher levels of heme and lipopolysaccharide biosynthesis. A relative abundance of *Bacteroides* species in donor FMT stools was associated with higher rates of remission in recipients, likely explained by antagonist interactions between *Bacteroides* and *Prevotella* species [54].

The beneficial effects of FMT in UC may also be mediated by other members of the intestinal microbiota, such as viruses and fungi. In a small cohort of nine UC patients, a numerical trend towards reduction in eukaryotic viral richness was observed in FMT responders compared with non-responders (*p* = 0.056) [55]. The pro-inflammatory role of Candida species was highlighted in a prospective trial of 39 patients. A relative abundance of Candida species pre-FMT was associated with increased bacterial diversity, which likely implies a microbiota more amenable to FMT engraftment. A reduction of Candida species post-FMT administration correlated with improved clinical and endoscopic outcomes; such an impact was not reproduced in patients receiving placebo [56]. A better understanding of the FMT-induced modifications of bacterial taxonomy, as well as trans-kingdom interactions, will hopefully improve the selection process of FMT donors and recipients, thus improving the overall efficacy of FMT in the management of active UC.

### 2.5. Dietary Studies in UC

An increasing number of studies have highlighted the importance of macronutrients in the aetiology of IBD [57,58,59], however data is limited regarding their impact on the gut microbiota. In one study that explored the role of animal-based diets, it was found that the increase in the abundance and activity of *Bilophila wadsworthia* on the animal-based diet group supports a link between dietary fat and the overgrowth of microbes able to prompt the host to IBD [60]. Table 1 highlights the published studies assessing the impact of diet in the incidence and treatment of UC.

#### 2.5.1. Low-FOPMAP Diet

A diet low in FODMAP (fermentable oligosaccharides, disaccharides, monosaccharides and polyols) may likely improve the quality of life of patients with irritable bowel syndrome (IBS) [61]. Data are scant regarding the effects of a diet low in FODMAPs in the clinical improvement of patients with IBD. An RCT was performed to investigate the effects of a low FODMAP diet on persistent gut symptoms, microbiome diversity and markers of inflammation in patients with quiescent IBD [62]. Patients following a low FODMAP diet (14/27, 52%) reported improved gut symptoms 4 weeks after following the dietary regime, compared to control diet (4/25, 16%, *p* = 0.007). Targeted stool samples analysis identified that patients following a low FODMAP diet had a significantly lower abundance of *Bifidobacterium adolescentis*, *Bifidobacterium longum* and *Faecalibacterium prausnitzii* than patients on control diet with no differences in relation to microbiome diversity and markers of inflammation [63].

#### 2.5.2. Specific Carbohydrate Diet

Specific carbohydrate diet (SCD) is a nutrition strategy that limits the consumption of certain carbohydrates. A retrospective paediatric study aimed to evaluate the potential effects of the SCD in 6 patients with active UC. The mean PUCAI for patients with active UC decreased from a baseline of 28.3 ± 10.3 to 20.0 ± 17.3 at 4 ± 2 week, to 18.3 ± 31.7 at 6 mo. Evidence shows that the SCD may be effective in decreasing disease activity and deserves further investigation and possible integration in the community but mechanisms involving the gut microbiome are lacking [64].

#### 2.5.3. Anti-Inflammatory Diet

The anti-inflammatory diet (IBD-AID) is a potential dietary therapy for IBD patients, restricting the ingestion of certain carbohydrates aiming to reduce symptoms and gut healing. A study recruiting forty patients with IBD were consecutively offered the IBD-AID to help treat their disease and were retrospectively reviewed. Of those forty adult patients, eleven were included in the final analysis and underwent further medical record review. After following the IBD-AID for at least four weeks, all patients were able to discontinue at least one of their prior IBD medications, and all patients had improvement of their symptoms, including reduced bowel frequency suggesting a potentially beneficial effect of this dietary strategy in clinical outcomes [65]. Objective measures of inflammation, such as faecal calprotectin or endoscopy, were not assessed. The impact of this diet on the gut microbiome is still not fully elucidated.

#### 2.5.4. High-Fibre Diet

Fibre ingestion can shape the structure of the gut microbiome and can contribute to colonic homeostasis, intestinal integrity, thus leading to a lower disease risk [66]. Resistant starch and pectin increase the relative abundance of butyrate-producing bacteria while reduction of dietary fibre consumptions is associated with a decrease in butyrate-producing bacteria such as *Faecalibacterium*, as well as an increase in mucus-eroding microbiota such as *Akkermansia muciniphila* and *Bacteroides caccae.* Dietary fibre is fermented in the colon and provide energy substrates for the colonocytes. Importantly, healthy subjects supplemented with fructooligosaccharides and galactooligosaccharides exhibit an increased abundance of Bifidobacteria and Lactobacilli [67].

#### 2.5.5. Mediterranean Diet

The Mediterranean diet (Md) is largely known by its inflammatory characteristics and cardiovascular benefits. A prospective study aimed to identify the impact of Md on the nutritional state, liver steatosis and clinical disease. The study reported a significant reduction of malnutrition-related parameters and improvement of liver steatosis was observed in 84 UC patients after short-term dietary intervention [58]. Specifically, they noticed that, after 6 months of a Mediterranean diet, fewer patients with UC had active disease (14 of 59 [23.7%] at the start of trial vs. 4 of 59 [6.8%] at 6 months following diet, *p* = 0.004) and furthermore, this study highlighted that a Md was associated with an increase in quality of life [68], thought to be related to reduced disease activity and lower body mass index. A trial from Henriquez Sanchez et al. also shows that adherence to a Md seems to correlate with higher physical and mental quality of life scores [69].

Increased intestinal permeability in IBD is thought to induce translocation of bacterial lipopolysaccharides into the portal system and cause low-grade liver inflammation, thus potentiating the risk of non-alcoholic fatty liver disease (NAFLD) [70]. Liver dysfunction from NAFLD progression may induce modifications of bile acid composition, thus further perpetuating intestinal dysbiosis and inflammation [71]. It has been suggested that a Mediterranean diet might be associated with modulation of the gut microbiome by increasing SCFA production and lowering the production of secondary bile acids, p cresols, ethanol and carbon dioxide [62]. This was shown to be associated with a reduction in fragility but, as of yet, mechanisms in ulcerative colitis are yet to be elucidated [72].

#### 2.5.6. Gluten-Free Diet

Gluten consists of proteins that are partially resistant to proteolytic digestion, being a major dietary component in wheat, rye and barley. Non-celiac gluten sensitivity and the associated use of a gluten-free diet (GFD) have been used as a dietary strategy to better control IBS symptoms and recently as a possible comprehensive tool to manage inflammatory bowel disease (IBD). In a study trying to analyse the effect of a GFD on gut inflammation, 65.6% of patients described an improvement of their gastrointestinal symptoms and 38.3% reported fewer or less severe IBD flares. As of yet, it is unclear how a GFD affects the gut microbiome and therefore further prospective studies into mechanisms of gluten sensitivity in IBD are warranted [73].

#### 2.5.7. Omega-3

Clinical studies show that omega-3 fatty acids may have a possible role in the treatment of IBD. Deficiency in essential fatty acids is commonly seen in IBD patients, and omega-3 fatty acids supplements may benefit patients through the inhibition of natural cytotoxicity (by changing arachidonic acid metabolites) and/or improving oxidative stress [74]. In a cohort of heathy middle-aged and elderly women, both total omega-3 and DHA serum levels were significantly correlated with microbiome alpha-diversity after adjusting for confounders. Some of the associations with gut bacterial operational taxonomic unit appear to be mediated by the abundance of the faecal metabolite N-carbamylglutamate. These data suggest a link between omega-3 circulating levels/intake and microbiome composition independent of dietary fibre intake, particularly with bacteria of the *Lachnospiraceae* family [75]. A prospective cohort study of 25.639 patients aged 45 or above required participants to complete a validated seven-day food diary, of which food items were electronically converted into nutrients [76]. These patients were subsequently monitored for the development of UC: 22 incident cases were diagnosed over a median time from recruitment to diagnosis of 4.2 years. In these newly diagnosed cases, a higher dietary intake of docosahexanoic acid (DHA) was associated with a decreased risk of developing UC (OR = 0.47, 95% CI = 0.25–0.89, *p* = 0.02) despite adjusting for cigarette smoking, total energy intake and other fatty acids, which can alter the metabolism of DHA. A numerical trend of borderline statistical significance showed lower incident cases of UC in patients with increased dietary intake of total n-3 polyunsaturated fatty acids (PUFA) and eicosapentaenoic acid (EPA). EPA and n-3 PUFA may exert anti-inflammatory effects through selective metabolism to leukotriene B_5_, thought to be less potent as a leucocyte chemotactic agent compared to leukotriene B_4_, which is derived from n-6 PUFA [77]. These omega-3 fatty acids are also believed to promote the release of phospholipases D and inhibit protein kinase C, thereby hampering different inflammatory pathways^,^ [76]. More studies assessing the effect of omega-3 supplementation on the microbiota of IBD patients are required.

#### 2.5.8. Curcumin

Polyphenols, which constitute the active substances found in many plants, seem to have positive effects in the management of IBD via down-regulation of inflammatory cytokines and enzymes, enhancing antioxidant defence and suppressing inflammatory pathways and the cellular signalling mechanisms [78]. Their specific role in the microbiome and ulcerative colitis remains unclear.

## 3. Therapies Involving Microbial Manipulation in Pouchitis

### 3.1. Prebiotic Studies in Pouchitis

Welters et al. evaluated the effects of enteral inulin on ileo-anal pouch functioning by studying epithelial gene expression, cell turnover and mucosal morphology. Twenty patients were given 24 g of inulin daily for three weeks, then a four-week wash-out period, and a placebo for three weeks. Inulin supplementation did not significantly alter pouch mucosal functioning because neither epithelial homeostasis nor epithelial gene expression was significantly altered; however, the author concluded that enteric supplementation with 24 g/day of inulin led to a decrease of inflammation-associated factors, with an increase in butyrate production, decrease of secondary bile acids and significant decrease in the endoscopic and histologic pouch disease activity index score [79]. This is yet to be linked to the role in the microbiome but could provide some interesting mechanistics as to the importance of prebiotics in pouch integrity.

### 3.2. Probiotic Studies in Pouchitis

In a meta-analysis of the four RCTs, it was shown that probiotics have no effect on the maintenance of remission in pouchitis. However, when exploring individual studies, it has been demonstrated that 6 g/day of a multistrain probiotic containing a combination of lactic acid bacteria, streptococci and bifidobacteria can help maintain remission [80] and prevent acute pouchitis [81,82]. There is a single small trial suggestive that probiotics may be effective for acute pouchitis [83] but this was not replicated in a RCT [84] and hence its position in acute pouchitis remains uncertain.

When exploring some of the microbial mechanisms that changed during probiotic treatment of the pouch, Gionchetti et al. noted that faecal concentration of lactobacilli, bifidobacteria and *S. thermophilus* increased significantly from baseline levels only in the group receiving the multistrain probiotic containing a combination of lactic acid bacteria, streptococci and bifidobacteria (*p* < 0.01) [81]. The same group in a follow-up study found that the patients that benefited from the multistrain probiotic were associated with faecal colonization with probiotics [83]. In the other probiotic study reporting on changes in the gut microbiome, Kuisima et al. highlighted that a trial of Lactobacillus GG supplementation (10 LGG, 10 placebo) for 3 months changed the pouch microbiota, but was ineffective as primary therapy for a clinical or endoscopic response [85].

### 3.3. Antibiotic Studies in Pouchitis

Antibiotics remain the mainstay treatment for both acute and chronic pouchitis. In a meta-analysis it was highlighted that antibiotics could achieve remission in nearly three-quarters of cases [86]. However, these are based on a number of small randomised controlled trials and observational studies. When exploring the microbial changes that occur following antibiotics, the evidence is very heterogenous, meaning consistent signals are not yet found. Furthermore, the mechanisms that underpin the microbial changes that are responsible for the beneficial effect remain poorly understood. In one of the few mechanistic studies exploring the role of antibiotics and pouchitis, it was highlighted that antibiotics lead to an antibiotic-resistant microbiome to include abundance of facultative anaerobic bacteria genera to include *Escherichia, Enterococcus and Streptococcus*. This was associated with low virulence which helps to maintain remission [87]. This therefore suggests that antibiotics may be decreasing the virulence of the bacteria contributing to pouchitis.

### 3.4. FMT in Pouchitis

There have been a number of small studies that have explored the potential of faecal microbiota transplantation for chronic pouchitis. Overall, a meta-analysis highlighted that as yet there was a lack of evidence for its effectiveness in the treatment of chronic pouchitis [88]. The problem with many of these studies is the heterogeneity in study design, the methodology of delivering the faecal transplant, the relatively small number of patients and the variability of diagnosis of pouchitis. Interpretation of these studies remains challenging, especially understanding mechanisms that may underpin therapeutic success.

### 3.5. Future Directions in Pouchitis Management

Due to the small number of studies related to pouchitis, larger studies are needed to guide our understanding of future therapies to treat pouchitis. It is likely that a multicentre approach with mechanistic work that aims to understand the pathophysiology of pouchitis will help guide our treatment strategies and begin to personalise them.

## 4. Interventional Dietary Studies in Pouchitis

A longitudinal cohort study followed 172 patients within the first year after IPAA surgery investigating who developed pouchitis. There was a greater risk of pouchitis at 1 year in 13 patients with low fruit intake (30.8% for <1.45 servings fruit per day) compared with 26 patients with higher fruit intake (3.8% for >1.45 servings fruit per day). Additionally, higher fruit consumption correlated with increased microbial diversity and with higher abundance of various bacterial genera including *Lachnospira, Lactobacillus*, *Faecalibacterium* and *Ruminococcus* [89].

McLaughlin et al. studied the impact of exclusive elemental diet on the gastrointestinal microbiota and symptoms in patients with chronic pouchitis. In their case series, 7 patients with pouchitis following IPAA for UC were treated with exclusive elemental diet therapy for 4 weeks. The median stool frequency significantly decreased from 12 to 6 per day. However, there was no significant difference in quality-of-life scores or pouch disease activity index before and after treatment. There were no significant changes in the concentration of bacteria after treatment. There was a trend towards an increase in the concentration of *Clostridium coccoides* and *Eubacterium rectale*. The authors concluded that elemental diet therapy appeared to improve the symptoms of pouchitis in some patients but is not an effective strategy for inducing remission [90].

## 5. Conclusions

Current evidence of probiotics, FMT and dietary interventions in the treatment of UC is encouraging although significantly limited by the small sample size of published trials and ongoing uncertainties about the causality of dysbiosis in intestinal inflammation. Modulation of the gut microbiome may allow clinicians to personalise therapeutic regimens in UC, tackling inflammatory disorders by targeting microbes involved in the pathogenesis of disease. Adding concomitant microbial therapies to standard therapy may amplify the relatively modest clinical benefits of the current therapeutic armamentarium. Further RCTs are required to understand the potential risk/benefit profile of therapies involving microbial manipulation.

## Figures and Tables

**Figure 1 nutrients-13-01780-f001:**
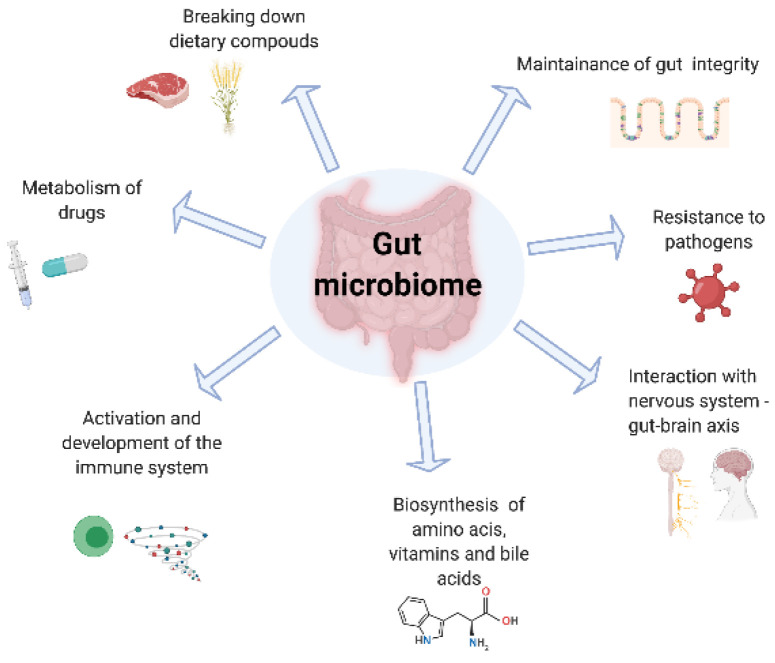
The role of the gut microbiome.

**Figure 2 nutrients-13-01780-f002:**
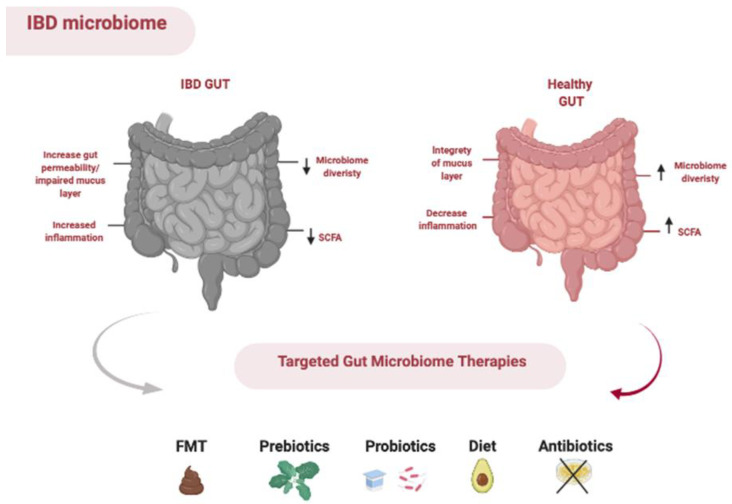
Methods of manipulating the gut microbiota.

**Table 1 nutrients-13-01780-t001:** Summary of published dietary studies regarding incidence and treatment of ulcerative colitis.

Authors and Date of Publication	Country	Studied Diets	Number of UC Patients	Study Design	Types of Outcomes
Li et al. 2015	Worldwide	Vegetable protein	896	Prospective	Incidence of UC
Dong et al. 2020	Worldwide	Animal protein	418	Prospective	Incidence of UC
Prince et al. 2016	United Kingdom	Low FODMAP diet	38	Prospective	Clinical response of UC
Cox et al. 2019	United Kingdom	Low FODMAP diet	26	Randomised, controlled trial	Clinical response of UC and microbiota composition changes
Obih et al. 2016	United States of America	Specific carbohydrate diet	6	Retrospective	Clinical response of UC
Olendzki et al. 2014	United States of America	Anti-inflammatory diet	3	Retrospective	Clinical response of UC
Chicco et al. 2021	Italy	Mediterranean diet	84	Prospective	Clinical response of UC
Herfarth et al. 2014	United States of America	Gluten-free diet	615	Cross-sectional, longitudinal	Clinical response of UC
John et al. 2010	United Kingdom	Omega-3 fatty acids	22	Prospective	Incidence of UC
Lu et al. 2017	Worldwide	Curcumin	244	Prospective	Incidence of UC

UC for ulcerative colitis, FODMAP for fermentable oligosaccharides, disaccharides, monosaccharides and polyols.

## Data Availability

No new data were created or analyzed in this study. Data sharing is not applicable to this article.

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
