# Peer review of "The Microbiome as a Therapy in Pouchitis and Ulcerative Colitis"

_nutrients, 2021, doi:10.3390/nu13061780_

Round 1

Reviewer 1 Report

The review was improved and many comments made by the reviewers were addressed. The text can be accepted for publication after just a few comments below. 

Minor details:

  • Below the table, remove the word "Legend".
  • Line 457, I would keep the word "management" instead of "incidence". 
  • I miss a conclusion on the pouchitis part of the story. 

Author Response

We thank you for your comments and hope that you accept the modifications detailed below based on your suggestions. 

  • Below the table, remove the word "Legend".
    Response: we removed the word "Legend". 
  • Line 457, I would keep the word "management" instead of "incidence". 
    Response: we changed it back to the "management".
  • I miss a conclusion on the pouchitis part of the story.
    Response: we added a concluding statement to the pouchitis section as seen on lines 631-635.

Reviewer 2 Report

It is a good review. I have some small changes: 

You need to define what is IBD, (Crohn's and colitis….), before using the words with Colitis.

Line 109: remove one of the maintenance of

Line 235: E. Coli Nissle 1917, is E. coli Nissle 1917

Line 528: is this: 25,639 or 25.639?

Author Response

We thank you for your comments and hope that you accept the modifications detailed below based on your suggestions. 

  • You need to define what is IBD, (Crohn's and colitis….), before using the words with Colitis.
    Response: we modified the Introduction section in order to clarify the definitions of IBD, Crohn’s disease and UC.
  • Line 109: remove one of the maintenance of
    Response: we removed the redundant "maintenance of".
  • Line 235: E. Coli Nissle 1917, is E. coli Nissle 1917
    Response: we corrected the typo.
  • Line 528: is this: 25,639 or 25.639?
    Response: we changed it to the European format of using a period for thousands, instead of a comma.

Reviewer 3 Report

The manuscript titled “The microbiome as a therapy in pouchitis and ulcerative colitis” by Jean-Frédéric LeBlanc and coworkers, they have described the knowledge of gut microbiome therapies in ulcerative colitis and pouchitis. I have few concerns regarding the present manuscript

-Thank you for the opportunity to revise the present manuscript, I noted that is a revised manuscript. However, the references are not in the line with Nutrients’ guidelines.
-Line 22, a missing abbreviation for ulcerative colitis
-Why is the main reason to select UC and pouchitis and not Crohn’s disease.
-In the introduction section the authors need to define microbiome and also microbiota, seems that authors have used these words as synonyms. The last paragraph of the introduction is important to explain the role of gut microbiota in UC, however, more information is required.
-¿Why the microbiota is important in UC? is important to explain with more important information the topic 1.1.
-Again, intestinal signatures for IBD are required in the main document to discuss treatment for UC or pouchitis
-Why the authors add information about antibiotics?
-Negative results are missing in the present document
-Thank you to the authors, the diet topic is well-written and discussed
-Why pouchitis results are not defined as a table? 

Author Response

We thank you for your comments and hope that you may accept the modifications detailed below based on your suggstions. 

  • Thank you for the opportunity to revise the present manuscript, I noted that is a revised manuscript. However, the references are not in the line with Nutrients’ guidelines.

Response: we discussed this issue with Ms. Bheryl Zhang, who mentioned that the Nutrients production team would perform the editing changes should the manuscript be accepted.

  • Line 22, a missing abbreviation for ulcerative colitis
    Response: we corrected the typo.
  • Why is the main reason to select UC and pouchitis and not Crohn’s disease.
    Response: we were invited to write a targeted review on microbial therapies in UC and pouchitis. We modified the Introduction section to clarify the definitions of UC and Crohn’s disease.
  • In the introduction section the authors need to define microbiome and also microbiota, seems that authors have used these words as synonyms. The last paragraph of the introduction is important to explain the role of gut microbiota in UC, however, more information is required.
    Response: we agreed with your suggestion and added clearer definitions for microbiome and microbiota, as seen in lines 26 and 27, as well as ensured that the correct term was used throughout the manuscript. We added more information regarding the pathophysiology of the gut microbiota, as seen in lines 117 to 119 and lines 150 to 157.
  • ¿Why the microbiota is important in UC? is important to explain with more important information the topic 1.1.
    Response: we added more information regarding the pathophysiology of the gut microbiota, as seen in lines 117 to 119 and lines 150 to 157.
  • Again, intestinal signatures for IBD are required in the main document to discuss treatment for UC or pouchitis
    Response: we clarified the importance of intestinal microbial signatures in section 1.1.
  • Why the authors add information about antibiotics?
    Response: we believe that microbial manipulation with antibiotics is a relevant topic for this review summarising the existing therapies of UC and pouchitis. It was also suggested by one of the reviewers to expand this section, as was completed in the previous revision.
  • Negative results are missing in the present document
    Response: we believe that we included more than a few negative trials in the probiotic, antibiotic, FMT and dietary sections. We had to remove previously-included studies in view of the fact that we reached the maximal word count proposed by the Nutrients journal.
  • Thank you to the authors, the diet topic is well-written and discussed
  • Why pouchitis results are not defined as a table? 
    Response: we believe that such a table would add little to the manuscript in view of the small number of trials performed and the heterogeneity of their methodology.

Round 2

Reviewer 3 Report

Thank you to the authors for taking into account my previous comments. The manuscript now is improved

This manuscript is a resubmission of an earlier submission. The following is a list of the peer review reports and author responses from that submission.

Round 1

Reviewer 1 Report

The subject is interesting since most of the studies in the last decade point on that there is a link between the IBD/ UC disease relapses and the microbiota. Unfortunately, there are not so many clinical studies out there on the subject, with a sufficient number of patients included in the experiments. Therefore, it makes this study less credible. I miss a discussion section in this paper, where all the parameters that have been written from different studies, were discussed from the author's point of view. The author described many clinical studies in this paper, but without writing the number of the patients involved in each study. It is important for the readers to know how many participants were in each study.  Please state that in each clinical study writing in this paper, especially for FMT study.

Questions to the author:

  1. Line 48: What do you mean by genetic predisposition to inflammation?
  2. Line 49: Not all the pouch patients experience pouchitis, will you please specify?
  3. Line 85-88: Please refer to reference? Where does that statement come from?
  4. Line 145: in another study, E. coli Nissle, had a negative effect on the patient remission outcome. Please refer to that study: Andreas Munk Petersen, June 2004, Gastroenterology 1016/j.crohns.2014.06.001
  5. Line 150-162, Please refer to the original paper with the clinical experiment. However, this study is too small to draw such a conclusion. 10 patients in the Bifidobacterium group and 9 patients in the placebo group is statistically not enough. This has to be stated in your paper.
  6. Line 455-458: Do you believe that antibiotic treatment has a beneficial effect and makes the bacterial less virulent or is it because it diminished most of the bacteria at the area of the inflammation and that is the reason the patient achieves remission after antibiotic therapy? Most of the time patients after antibiotic therapy will have disease relapses. When it comes to the paper you are referring to, what kind of virulent gene and which kind of bacteria they have been investigating? This needs to be discussed in your paper.

Reviewer 2 Report

[Nutrients] Manuscript ID: nutrients-1145520

Title: The microbiome as a therapy in pouchitis and ulcerative colitis

The submitted review paper discuss the importance of the microbiota of ulcerative colitis and patients with pouchitis. It also discusses the use of prebiotics and probiotics to control both inflammatory processes as well as other components of the diet.

Major comments:

Abstract

The content of the abstract is superficial and needs improvement. It also does not match the topics in the text, since a lot of diet-related information is added to discuss improvement of disease states (either ulcerative colitis or pouchitis).

Introduction

  • The first two paragraphs (Lines 19 to 29) are disconnected and again gives only general information. My suggestion is to add more information about bacterial communities that colonize the gut and can be associated with healthy and diseased states.
  • More detail on Ulcerative Colitis is needed. Please discuss the unknown etiology of the disease and what is known about its development. How does the microbiota impact this disease and how can it be used to improve disease? Maybe also add some statistics of incidence and prevalence. I would also be interesting to demonstrate why microbiota would be a good option for treatment (what is available? Why do we need other strategies of treatment? Etc)
  • On line 40, the authors talk about a model of ulcerative colitis that allow the longitudinal evaluation of the microbiome changes from healthy to disease state. My understanding is that using microbiota prom patients who underwent ileoanal pouch to justify the longitudinal analysis of microbiota is controversial. A patient who has this procedure has a history of inflammatory bowel disease, which can indicate that these patients already have an altered microbiota compared to healthy controls. In addition to that, the inflammation called pouchitis can also have effect on the microbiome (e.g. increasing in pathological strains vs beneficial strain of bacteria). Therefore, evaluation of pouchitis does not seem to be ideal model to evaluate changes from healthy to disease state as the authors are proposing. My suggestion would be to focus on longitudinal studies of patients with UC regarding the microbiome changes throughout disease state (active periods vs remission). Longitudinal feces samples from patients can also be collected overtime.

See the following papers:

  • Schirmer M, Denson L, Vlamakis H, Franzosa EA, Thomas S, Gotman NM, Rufo P, Baker SS, Sauer C, Markowitz J, Pfefferkorn M, Oliva-Hemker M, Rosh J, Otley A, Boyle B, Mack D, Baldassano R, Keljo D, LeLeiko N, Heyman M, Griffiths A, Patel AS, Noe J, Kugathasan S, Walters T, Huttenhower C, Hyams J, Xavier RJ. Compositional and Temporal Changes in the Gut Microbiome of Pediatric Ulcerative Colitis Patients Are Linked to Disease Course. Cell Host Microbe. 2018 Oct 10;24(4):600-610.e4. doi: 10.1016/j.chom.2018.09.009. PMID: 30308161; PMCID: PMC6277984.
  • Llewellyn SR, Britton GJ, Contijoch EJ, Vennaro OH, Mortha A, Colombel JF, Grinspan A, Clemente JC, Merad M, Faith JJ. Interactions Between Diet and the Intestinal Microbiota Alter Intestinal Permeability and Colitis Severity in Mice. 2018 Mar;154(4):1037-1046.e2. doi: 10.1053/j.gastro.2017.11.030. Epub 2017 Nov 23. PMID: 29174952; PMCID: PMC5847454.
  • Lines 46 to 52: This paragraph is confused and lacks a lot of essential information for the reader. If focusing on ileoanal pouch and pouchitis, please explain: 1. What is ileoanal pouch and when is this performed? 2. What is pouchitis? 3. What is the association between microbiota and pouchitis? 4. Why do patients with colitis have to undergo this procedure?

Section 1.1

  • Line 55: change the title of this section – it is not related to the text
  • Line 57: Same issue of the introduction – superficial information, need to develop more. The authors are discussing that data from NGS have improved our understanding of gut microbiome but does not mention specific information about any study.
  • Line 61 to 68:
    • Explain the inflammatory process favoring the epithelial barrier integrity (e.g. tissue regeneration)
    • Role of microbiota in this process and how microbiota enters the mucosa causing inflammation
    • Role of mucus in protecting the mucosa (why is it important? Which cells produces it? What happens to mucus production in UC?) It is important to also correlate with the state of chronic inflammation seen in long-standing UC, in which loss of goblet cells is observed.
    • Line 73: the text needs a final sentence.
    • Figure 1: use this figure to complement the information in the text and add an explanatory legend.

Section 2.4

  • Because the text lacked information in the beginning regarding the types of bacteria that are good or bad for disease, it is hard to understand why the bacteria cited are affecting or protecting disease state. It would also be interesting to have more discussion on why markers such as SCFA and secondary bile acids, heme and LPS are important to measure healthy/disease state.

Section 2.5 and 2.6

  • This section is extremely important for IBD. However, I could not see a link between diet and microbiota in the text since most of the studies cited does now find any correlation with the microbiota.
  • For the Mediterranean diet, please confirm if the diet is known to be anti-inflammatory or inflammatory (Line 279).
  • Please explain the controversial data that suggests an improvement in quality of life for these patients versus the increase in liver steatosis for these patients.
  • My suggestion would be to generate a table with the data from 2.5 and 2.6.

Section 3

  • This section should be reviewed based on the comments made about the model. There needs to be more information about the surgical procedure and how the inflammatory process starts in the local of surgery, if the author decide to keep it. Because the model does not seem strong to support the proposal, I would remove this section from the review.

Minor comments

  • Make sure all the abbreviations are explained through the text at least one time (e.g. RCT, IPAA, FFQ).
  • Confirm that all the bacteria names are written in the proper way.

Reviewer 3 Report

The authors nicely summarise the Gut Microbiota of UC patient and its important as a therapy in pouchitis and UC. I would like to share with some comments that I believe it will help to manuscript sounds better and more complete:

  • I would rather change “our human gut microbiome” to “human gut microbiome” or “our microbiome” to “human microbiome”
  • It will be nice if the authors introduce the ileoanal pouch (or pouchitis)for the readers who are not from this field?
  • The authors can have a look at the Macpherson’s group 2019 manuscipt where they showed the consistent and non-consistent findings of UC microbiota profile. This study can help them to extend this paragraph.
  • Bacteria name should be in italic. These should be corrected through all the text.
  • Uncovering microbiome-based therapies through mechanistic understanding of pathogenesis in pouchitis section seems more basic information on pouchitis. Title can be changed something like microbial profile in patients withh pouchitis.
  • If one just Google for antibiotics in UC, you can come across with this reeview: https://www.ncbi.nlm.nih.gov/pmc/articles/PMC4716021/ as you can see many studies are mentioned in this manuscript. I’d suggest to the author to extend this section.
  • Same for dietary studies…
  • Conclusion is too short and I believe it can be rather extended to cover up the messages that authors want to share with scientific society.